# The effects of free trade agreements on the stock market: Evidence from Vietnam

**Huy Pham**[1][©], **Priyantha Mudalige**[2][©], **Hanh Le**[1][©], **Mai Bui**[1][©], **Van Nguyen**[3][©], **Vikash Ramiah**[4][©], **Tuan Chu**[1][©], **Tuan Hung Vu**[5]*

**1** The Business School, RMIT University, Ho Chi Minh, Vietnam, **2** Faculty of Science, Agriculture, Business and Law, University of New England Business School, Armidale, Australia, **3** Faculty of Finance and Banking, Ton Duc Thang University, Ho Chi Minh, Vietnam, **4** Faculty of Business, University of Wollongong, Dubai, UAE, **5** The Business School, Bournemouth University, Poole, United Kingdom

© These authors contributed equally to this work.

* tvu@bournemouth.ac.uk

**Data Availability Statement:** Data cannot be shared publicly because of the copyright. Data are available from the Thomson Reuter Eikon DataStream (contact via https://eikon.refinitiv.com/

## Abstract

This study examines the effects of news events related to the European Union-Vietnam Free Trade Agreement (EVFTA) on the Vietnam stock market from 2010 to 2020. We calculate sectoral abnormal returns prior to, during, and after announcements and find that the Vietnamese stock market is susceptible to these events. We discovered that the announcement had a negative impact on the market, which might diminish the effectiveness of the Agreement. The findings show that more than half of Vietnam's sectors had an immediate reaction to EVFTA announcements, with fourteen reacting negatively and six responding positively. Two of the ten events did not have any immediate impact on these industries but all events resulted in either early or delayed reactions. We also find market scepticism and major changes in the deal led to the emergence of a diamond risk structure. We run multiple robustness tests to account for market integration and other factors that may affect stock returns. In addition, we explore potential sectoral systematic risk changes following these occurrences using different ARCH-type models. These additional tests confirm the robustness of our findings.

## 1. Introduction

Vietnam joined the Association of South-East Asian Nations in 1995, opening its gates afterwards for participating in the Free Trade Agreements (FTAs) with regional and international partners. By 2022, Vietnam has signed 15 FTAs with different regions and countries, such as China, Korea, Japan, India, Australia, New Zealand, etc. These FTAs are believed to bring both opportunities and challenges to Vietnam. On 30 June 2019, the European Union Trade Commissioner Cecilia Malmstrom and Vietnam's Minister of Industry and Trade Tran Tuan Anh signed the European Union-Vietnam Free Trade Agreement (EVFTA), described as "the most important ambitious free trade deal ever concluded with a developing country" (Accessed on 6 November 2019: https://www.reuters.com/article/us-eu-vietnam-trade/vietnam-eu-sign-landmark-free-trade-deal-idUSKCN1TV0CJ) [1]. The deal was signed three-and-a-half years after negotiations ended in December 2015, and it is expected to eliminate up to 99% of tariffs

) for researchers who meet the criteria for access to confidential data. The data underlying the results presented in the study are available from Thomson Reuter Eikon DataStream.

**Funding:** The authors received no specific funding for this work.

**Competing interests:** The authors have declared that no competing interests exist.

between the European Union (EU) and Vietnam. The European Union accounts for 11.6% (equivalent to US$55.8 billion) of Vietnam's trade value (Accessed on 6 November 2019: https://customsnews.vn/vietnams-10-biggest-trading-partners-9621.html), and they expect that the deal will further boost trade and investment between the EU and Vietnam, which has quintupled over the last ten years(Accessed on 6 November 2019: http://www.europarl.europa.eu/legislative-train/theme-a-balanced-and-progressive-trade-policy-to-harness-globalisation/file-eu-vietnam-fta). Although some tariffs will be cut over a 10-year period and agricultural product exports will be limited by quotas, Vietnam expects to add €15 billion a year of additional exports to the EU by 2035. The EVFTA covers provisions on non-trade barriers, competition policy, as well as public procurement, among other things [2]. Moreover, the EVFTA will be an opportunity to increase exports significantly for Vietnam as Vietnam is the EU's 15[th] trade-in goods partner globally and the largest trading partner in the Association of Southeast Asian Nations (ASEAN). Additionally, the EU is one of the largest foreign investors in Vietnam (Accessed on 6 November 2019: https://policy.trade.ec.europa.eu/eu-trade-relationships-country-and-region/countries-and-regions/vietnam_en). Therefore, the EVFTA is expected to impact the Vietnamese sectors significantly. Apart from the benefits, Vietnamese firms may also face problems such as the lack of competence and management experiences and low production capacity, especially in high technology industries. A report by [3] indicated that less than 50% of Vietnam enterprises could take advantage of FTA. Hence, investors may have different expectations regarding the impacts of FTA on Vietnamese firms.

The impacts of FTA have been documented in the literature; however, the evidence is inconclusive. [4], for instance, find positive impacts of an East Asia FTA on GDP and welfare in member countries, while the effects on non-members are negative. Other FTAs, such as the North American Free Trade Agreement (NAFTA, 1994), ASEAN–India FTA (2010), EU-Korea FTA (2008), and Southern African Development Community free trade (2008), brought positive results to the whole economy of both parties, such as the increase in the trading balance as well as GDP. The results found that NAFTA, for example, positively affected the US and Mexico stock markets; however, the magnitude is higher for Mexico's economy. Although a large body of research focuses on the economic effects of free trade agreements, few studies investigate the impact of FTA on the stock market. The literature shows mixed evidence whereby some industries experience positive returns while others do not. For instance, [5, 6] find that free trade agreement-related news announcements positively impact stock markets. [7] also show the positive effects of the U.S.-Singapore free trade agreement announcements on the Singapore stock market. The stock market can also be an unbiased barometer to evaluate economic policies and their effects, particularly on allocative efficiency [8, 9] and investor perceptions [10]. Some other studies examine the relationship between FTAs and stock return and find inconclusive results on any direct relationship [11].

In the Vietnam context, several studies investigated the economic impacts of EVFTA at the sectoral level, such as [12, 13]; however, no prior study explicitly examines the effects of EVFTA-related announcements on the Vietnam stock market. The Vietnam stock market is unique since it is still considered a frontier market despite robust economic development. Vietnam is one of a few countries that top the global economic growth, and the ASEAN-5 and the EVFTA are expected to help Vietnam maintain its economic growth in the long term. In addition, the current literature on stock market reaction to FTA-related news announcements may not be generalisable to the Vietnamese market due to its unique nature, investor perspectives and regulatory environment. Hence, Vietnam is an ideal testing ground to examine the impacts of FTAs on a frontier stock market.

The potential impact of the EVFTA on the Vietnam stock market holds significant economic importance for several reasons: (i) The Vietnam stock market is a frontier stock market,

and the market's reaction to the EVFTA-related news might be attributed to investors' collective stance on global market exposure. It may respond to economic openness, which will be useful to investors and policymakers. (ii) In addition to the overall market impact, the effect of the FTA can differ by industry sector. Some industries will profit from lower tariffs and more market access, while others will confront increased competition and poor economic performance. (iii) This trade agreement can potentially affect macro and firm-level risk and economic and firm-level performance. Hence, this discovery will be useful to policymakers. (iv) This research contributes to the understanding of informed trading by providing crucial insights on informational efficiency around the announcement on the Vietnam stock market.

We employ an event study methodology that tracks the market reaction (as proxied by abnormal stock returns) to closely examine how the Vietnam stock market reacts to events leading to the EVFTA. In addition, we check the robustness of our findings by conducting a series of tests, including nonparametric ranking test by [14], conditional distribution approach by [15], supplemented by the use of the Fama-French (2015) five-factor model by [16] and market integration approach. Although the EVFTA will benefit Vietnam in many ways, we anticipate that some Vietnamese industries will suffer competition due to the agreement, making it more straightforward for EU companies to operate in Vietnam. Therefore, we expect several changes in the systematic risk of Vietnamese sectors. We use various ARCH models to examine these changes following the events around the EVFTA. In summary, we find that various Vietnamese industries have suffered due to the EFVTA events. Our results also show that the Vietnam stock market is highly sensitive to these announcements. We also find evidence for a diamond risk structure arising when EVFTA is approved and agreed upon by Vietnam and the EU Parliament.

The remainder of the paper is structured as follows. Section 2 presents a literature review on the free trade agreement and its impact on the stock market around the world. Section 3 describes the methodology used in this study. Section 4 discusses the empirical findings, and Section 5 concludes the paper.

## 2. Literature review

Free Trade Agreements (FTAs) are forms of trade pacts between governments that eliminate tariffs, quotas and other barriers for a variety of items traded between involved parties [17]. Several factors contribute to countries embarking on free trade agreements. [18] identify several reasons behind free trade agreements: promote economic growth, comparative advantage and economies of scale, reduce uncertainty to market access, prevent future protectionism, recognition, credibility and continuity, and reduce the trade deficit.

### 2.1 The impact of FTAs on stock market returns

The relationship between FTAs and stock market performance has been the subject of extensive academic scrutiny. While the economic implications of FTAs on trade volumes, GDP, and industrial output are well-documented, their immediate and residual effects on stock markets present a complex narrative. [9] positioned the stock market as an arbiter for assessing economic policy decisions, particularly those revolving around FTAs. According to [9], the stock market's dual role–as an unbiased evaluator and a source of market data for event studies–offers a real-time reflection of investor sentiment and expectations concerning FTAs. This perspective suggests that stock market reactions can serve as immediate barometers of the perceived success or failure of FTAs. [11] conducted a seminal study on the effects of the US-Morocco free trade agreement on the Casablanca Stock Exchange. Their findings revealed that the agreement, which eradicated duties on over 95% of goods and services, had a pronounced positive influence on stock returns. These results reinforce the notion that stock

markets, as forward-looking entities, adjust swiftly to the anticipated economic benefits of FTAs. [19] examine the relationship between FTAs and the volatility of stock returns and exchange rates, focusing on the North American Free Trade Agreement (NAFTA). The study finds that the implementation of NAFTA has reduced the volatility of stock returns and exchange rates. These results suggest that FTAs can contribute to a more stable and predictable stock market environment, which can benefit investors.

Furthermore, [20] investigate the economic effects of Brexit on the stock market. The study analyses stock price reactions to the Brexit referendum and finds evidence supporting the specific factors model of international trade. These findings imply that changes in trade policy, such as the establishment or modification of FTAs, can have a significant impact on stock market performance. Positive expectations regarding the potential benefits of FTAs can drive stock prices higher. Besides, [8, 21] suggested that governments in developing countries, such as Vietnam, could bolster investor confidence and realize economic gains lead to stock market development from FTAs by forming agreements with developed, stable partners and leveraging their comparative advantages in labour-intensive trades and production, respectively. Based on the preceding discussion, our first hypothesis is formed as follows.

**H1**: EVFTA-related events significantly affect the Vietnam stock market.

## 2.2 The impact of FTA news and events on sectoral stock returns

The profound influence of FTA news and events on stock markets, particularly within specific industry sectors, has garnered substantial attention in academic research. The literature typically underscores the flow effects of FTA announcements and implementations on various sectors, substantiating that these agreements and their associated news generate discernible and often statistically significant market responses. [6] illuminated the differential impacts of NAFTA on various sectors within the Mexican Stock Exchange. Positive abnormal returns were notably observed on the approval and negotiation conclusion dates, with pronounced effects on industries like paper/cellulose, iron/steel, and electronics. Contrastingly, sectors such as insurance, chemicals, and mining demonstrated a weaker response, illustrating the heterogeneous effects of FTA news across different industries. [5] probed into the US-Canada FTA's impact on diverse sectors, revealing that events related to the agreement precipitated abnormal risk-adjusted price changes, signalling a tangible market impact. The research highlighted the nuanced sensitivities of various industries, such as textiles and computers, to the agreement, emphasizing that larger firms, particularly in the oil and gas, and computer industries, were more perceptibly affected by the FTA news.

Furthermore, [7] explored the US-Singapore FTA's impact on firm values listed on the Singapore Exchange, uncovering a generally positive market response to the agreement. However, the firm value increment was notably larger for the basic materials and healthcare industries, while some sectors, like consumer goods and technology, did not experience significant changes, signifying the sector-dependent nature of FTA effects. The literature reflects a multifaceted interplay between FTA news and industry-specific stock market responses, underscoring the critical influence of factors like firm size, industry type, and economic context. The findings collectively highlight the importance of considering sectoral nuances when examining the broader economic impacts of FTAs. This paper aims to fill this gap by examining the intricate dynamics between EVFTA news, events, and sector-specific stock market responses in an emerging country like Vietnam. The study, therefore, proposes the following hypothesis.

**H2:** The effects of EVFTA-related events vary across Vietnamese sectors.

### 2.3 The impact of FTAs on systematic risk

The intersection of FTAs and market risk introduces a complex paradigm in the global financial landscape, affecting diverse industries in various ways. Previous studies have explored how FTAs introduce a new level of market risk, impacting industries and stock markets. [19] offered insights into the stabilizing effects of FTAs on stock and foreign exchange markets by examining the Canada-US FTA and NAFTA. They discovered that while these agreements reduced volatility in US and Canadian stock markets, NAFTA increased volatility in the Mexican equity market, signifying that FTAs can affect member countries differently. [22] noted that factors such as return volatility, correlation, and domestic market performance play pivotal roles in determining per-unit-of-risk diversification gains for investors, with increased return volatilities and poor performance diminishing diversification benefits post-NAFTA. [23] also examined a sample of Mexican and Canadian stocks listed on the New York Stock Exchange and whether these stocks outperform the S&P 500, affecting NAFTA 1994. The results exhibit Mexican cross-listed stocks with negative cumulative excess returns. The authors suggest the results may be due to several factors, including asymmetric information, higher risk, and uncertainty of firms. However, Canadian stocks outperform the market, suggesting that these stocks have a NAFTA-related wealth effect. Trade liberalization resulting from FTAs stimulates varied responses across different sectors. [24] indicated that trade liberalization can have divergent impacts on exporting and non-exporting firms, with the former benefiting from reduced tariffs and enhanced foreign market access, while the latter may experience diminished profits due to intensified competition. Investigating the Gulf Corporation Council markets, [25] found that capital market openness significantly curtails total and idiosyncratic return volatility, with volatility decreasing as the stock market develops. This finding is further corroborated by [26], who found that market liberalization in China mitigated informed trading severity and augmented informational efficiency in the Chinese stock market. [27] also underscored the importance of systematic risk exposure in pricing stocks within trading blocks. Similarly, [28] elucidated how the liberalization and privatization of the Vietnamese Stock Market augmented the role of systematic risk over idiosyncratic risk in influencing the risk-return trade-off, allowing investors to achieve improved returns through portfolio diversification. These studies reflect a crucial acknowledgement of the pervasive influence of systematic risk following FTAs and market liberalisation.

The literature reveals that FTAs and subsequent market liberalisation introduce varied levels of market risk across different industries and countries, mediated by factors such as systematic risk exposure, diversification benefits, market volatility, and the specific economic and financial contexts of the involved nations. Understanding these multifaceted dynamics is crucial for navigating the complexities introduced by FTAs and for developing strategic policy and investment approaches that can harness the opportunities and mitigate the risks presented by such agreements. Therefore, this study will investigate the effects of EVFTA announcements on the systematic risk in Vietnam, an emerging country currently missing in the literature. Our third hypothesis is as follows.

**H3:** EVFTA announcements have different impacts on systematic risk across the sectors in Vietnam.

## 3. Methodology

### 3.1 Measuring the effects of the events

[29] propose the most recent development in event study methodology to measure the full value effect of an event for firms with traded options. This method is a generalisation of earlier

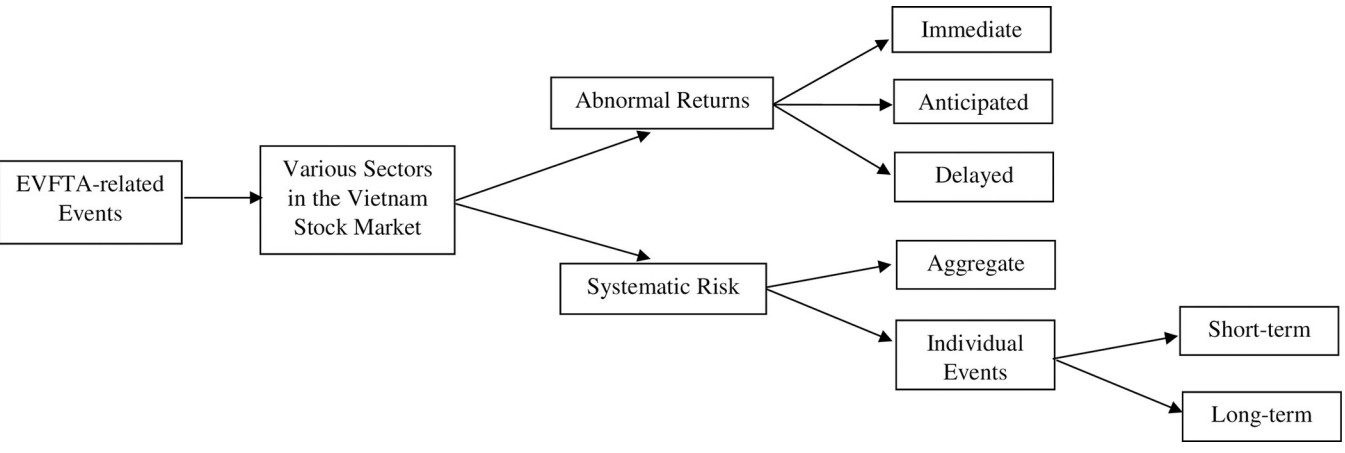

**Fig 1. Mechanism diagram for the impact of EVFTA-related events on the Vietnam stock market.**

work by [30–32] to unravel the value effects as a consequence of a merger announcement. The proposition about the usefulness of good estimates for event probabilities can be traced back to [33], who suggests that the change in stock price is due to partly anticipated events, and the estimates must be adjusted to obtain a proper measure of the full value effect of an event. [34–40] suggest that firm-specific attributes can be used to estimate the *ex-ante* event probability. The potential problem with this approach is that data on relevant firm-specific attributes may be scarce. Another common criticism of event study methodology is its inability to show the full effects of the events. [29], for instance, present a model to identify the unknown parameters that can be used to determine the full effect using stock and option prices. Although this approach provides several advantages, its implementation in this study is problematic because it is only appropriate for firms with traded options. Since our research is conducted on a comparative basis, even partial effects will do. Fig 1 presents a mechanism diagram to reflect the unique impact of EVFTA on the Vietnam stock market.

Recently, [41] examined the effects of information disclosure announcements on the Vietnam stock exchange by modifying the event methodology proposed by [42] and found statistically significant results around these events. To this extent, we follow [41] to examine the effects of ten key events around the EVFTA on the Vietnam stock market. Prior empirical studies document that news events influence stock prices ([43–47]), and good news tends to result in positive abnormal returns, and bad news leads to negative abnormal returns [44]. However, [48, 49] also point out that the events do not affect all sectors. Hence, we hypothesise favourable (unfavourable) events for a particular sector lead to positive (negative) abnormal returns. If an event is not newsworthy or does not influence a sector significantly, then the abnormal sectoral return is zero or statistically insignificant. These three possibilities are presented as follows:

$$\frac{\sum_s DAR_{it}}{N} < 0, \tag{1}$$

$$\frac{\sum_s DAR_{it}}{N} = 0, \tag{2}$$

where

$$DAR_{it} = In\left(\frac{PI_{it}}{PI_{it-1}}\right) - \beta_{it}^0 + \beta_{it}^1\left(r_{mkt} - r_f\right). \tag{3}$$

$\frac{\sum_s DAR_{it}}{N}$ represents the daily abnormal return of the sectors, $DAR_{it}$ daily abnormal return of firm $i$ at time $t$, $N$ is the number of firms within the sector, $PI_{it}$ is the stock price of firm $i$ at time $t$, $\beta_{it}^0$ and $\beta_{it}^1$ are the intercept and the slope of the CAPM model, respectively, $r_{mkt}$ is the market index of the Vietnam stock market and $r_f$ is the risk-free rate.

We calculate the abnormal returns five days before and five days after the events to capture the potential latency effects. The cumulative abnormal returns around the events are measured as follows:

$$CAR(d)_{it} = \sum_{n=1}^{d} DAR_{st+n} \qquad (4)$$

$$CAR(d)_{it} = \sum_{n=-1}^{d'} DAR_{st+n} \qquad (5)$$

where $CAR(d)_{st}$ is the cumulative abnormal return of a five-day period after the event of sector $s$ at time $t$. $CAR(d')_{st}$ is the cumulative abnormal return of a five-day period before the event of sectors $s$ at time $t$. The t-statistic determines the statistical significance of the result for each announcement.

## 3.2 Robustness checks

In this section, we check the robustness of the results using several alternative models. First, we use Fama and French five-factor model to re-estimate the abnormal return around events. As in Eq (3), our main model estimates abnormal return only controlling for the market risk premium. This model may not consider other important factors that influence the returns given the capital market imperfections. To include other relevant risk factors as proposed by [16], we include size (SMB), value (HML), profitability (RMW) and investment (CMA). The estimation model for new abnormal returns is modified as follows:

$$DAR_{it} = ln\left(\frac{PI_{it}}{PI_{it-1}}\right) - \beta_{it}^0 + \beta_{it}^1\left(r_{mkt} - r_{ft}\right) + \beta_{it}^2(SMB) + \beta_{it}^3(HML) + \beta_{it}^4(RMW) + \beta_{it}^5(CMA) + \varepsilon_{it} \quad (6)$$

where $\beta_{it}^0$ is the intercept of the Fama-French five-factor model. The coefficients; $\beta_{it}^1, \beta_{it}^2, \beta_{it}^3, \beta_{it}^4$ and $\beta_{it}^5$ represents market risk premium, size, value, profitability and investment factors, respectively. $\varepsilon_{it}$ is the model's error term.

Second, we use nonparametric tests to check whether abnormal returns are misrepresented in the distribution since abnormal returns can be significantly larger around event days and relatively insignificant during non-event days [50]. As a result, the distribution of abnormal returns is potentially misrepresented, resulting in high kurtosis, positive skewness and non-normality. The standard t-statistic computation in baseline results might depict these potential characteristics of the distribution inheriting from standard errors. Therefore, we use the ranking test by [14] and a nonparametric conditional distribution by [15] to check for potential biases. Following the nonparametric ranking test, we convert abnormal returns into ranks over a 260-day window as follows:

$$K_{it} = rank\left(ln\left(\frac{PI_{it}}{PI_{it-1}}\right) - \beta_{it}^0 + \beta_{it}^1\left(r_{mkt} - r_{ft}\right)\right) \qquad (7)$$

where $K_{it}$, is the rank of each firm $i$ at time $t$. The 260-day period consists of 244 days before

and 15 days after the event dates. We calculate the expected average rank for each firm in the sector and then compare the rank of each firm with the expected average rank, which is presented by $\bar{K}_i$, at time $t$ calculated as follows:

$$\bar{K}_{it} = 0.5 + \frac{260 \; days}{2} = 130.5 \tag{8}$$

The t-statistic of the Corrado test is used to check whether the event has a statistically significant effect on each sector, and it is estimated as follows:

$$t_{Corrado} = \frac{\frac{1}{N}\sum_{i=1}^{N}(K_{it} - 130.5)}{stdev(\bar{K}_{it})} \tag{9}$$

where $stdev(\bar{K}_{it})$ is calculated as

$$stdev(\bar{K}_{it}) = \sqrt{\frac{1}{T}\sum_{t=1}^{260}\frac{1}{N^2}\sum(K_{it} - 130.5)^2} \tag{10}$$

where $stdev(\bar{K}_{it})$ is the standard deviation of the average rank.

The result of the nonparametric ranking test is valid from the statistical point of view since it addresses the nonnormality, especially when applied to skewed and/or leptokurtic distributions [51].

Third, we conduct another nonparametric conditional distribution as an alternative approach to the event study methodology to address the nonnormality of returns. We use the kernel regression technique, which does not assume any underlying distribution and checks the conditional cumulative probability of the return on a general index. If the conditional cumulative probability has a value of less than 0.05, we conclude that the event has an extreme effect on the market.

Fourth, the Vietnam stock market is highly integrated with the international stock markets. Therefore, asynchronicity, market integration and spillover effects potentially influence abnormal returns and should be controlled to make the results robust. For this purpose, we use three market risk premia representing Asia, Europe, and the U.S., given these markets' integration with Vietnam. The controls are incorporated through three market risk premia representing Asia ($\tilde{r}_{mkt}^{Asia} - \tilde{r}_{ft}^{Asia}$), Europe ($\tilde{r}_{mkt}^{Europe} - \tilde{r}_{ft}^{Europe}$) and the U.S. ($\tilde{r}_{mkt}^{US} - \tilde{r}_{ft}^{US}$) into the CAPM and re-estimate the abnormal returns. The following equation shows our re-estimation model:

$$DAR_{it} = ln\left(\frac{PI_{it}}{PI_{it-1}}\right) - [\beta_{it}^0 + \beta_{it}^1\left(r_{mkt} - r_{ft}\right)] + \beta_{it}^2\left(\tilde{r}_{mkt}^{Asia} - \tilde{r}_{ft}^{Asia}\right) + \beta_{it}^3\left(\tilde{r}_{mkt}^{Europe} - \tilde{r}_{ft}^{Europe}\right)$$
$$+ \beta_{it}^4\left(\tilde{r}_{mkt}^{US} - \tilde{r}_{ft}^{US}\right) + \varepsilon_{it} \tag{11}$$

where the coefficients, $\beta_{it}^2$, $\beta_{it}^3$ and $\beta_{it}^4$ represents market risk premia from Asia, Europe and the U.S., respectively. The error term of the model is $\varepsilon_{it}$. The significance of re-estimated results is determined using the standard t-statistic.

## 3.3 Estimate changes in systematic risk

The events related to the free trade agreement may influence systematic risk because agreements on tariffs and the liberalisation of imports and exports can have an impact on overall market risk. As a result, news events on the changes may either increase, decrease or have a neutral effect on market uncertainty. For example, exports from the European Union under agreed tax and tariff terms may affect substitution industries in Vietnam, leading to an increase

in uncertainty in those industry sectors. Similarly, several industry sectors may secure a broader market due to favourable trading terms under the agreement, which might result in a decrease in uncertainty of the sectors. In this section, we examine how systematic risk is affected by the free trade agreement. To test for changes in systematic risk, we follow [52] and include interaction variables in asset pricing models. This model captures the average change in risk resulting from the implementation of the free trade agreement. An aggregate dummy variable (*AD*), which represents the events, takes the value of one on the event date and zero otherwise. The interaction variable is calculated via the multiplication of the aggregate dummy variable by market risk premium, and the model is as follows:

$$\tilde{r}_{It} - \tilde{r}_{ft} = \beta_I^0 + \beta_I^1[\tilde{r}_{mt} - \tilde{r}_{ft}] + \beta_I^2[\tilde{r}_{mt} - \tilde{r}_{ft}]*AD_t + \beta_I^3 AD_t + \tilde{\varepsilon}_{it} \tag{12}$$

where $\tilde{r}_{It}$ is the sector *i*'s return at time *t*, $\tilde{r}_{ft}$ is the risk-free rate at time *t*, $\tilde{r}_{mt}$ is the market return at time *t*, *AD* is a dummy variable that assumes a value of one on the event date and zero otherwise, $\tilde{\varepsilon}_{it}$ is the error term, $\beta_I^0$, is the intercept term such that $E(\beta_I^0) = 0$, $\beta_I^1$ is the average short-term systematic risk of the industry, $\beta_I^2$ is a measure of systematic risk for each sector, and $\beta_I^3$ is a measure the change in the intercept of Eq (12). By estimating Eq (12), the aggregate effect of the events on the stock market can be calculated.

The effects of opposite outcomes from different events may cancel out each other, which is a problem that can be dealt with by introducing an individual dummy variable (*ID*) for each announcement, taking a value of one on the event date and zero otherwise. By doing that, it becomes possible to identify the exact contribution of each event. Short-term changes in systematic risk following the implementation of the free trade agreement can be captured by the coefficients on interaction variables, which are obtained by multiplying each dummy variable by the market risk premium. In this case, we have

$$\tilde{r}_{It} - \tilde{r}_{ft} = \beta_I^0 + \beta_I^1[\tilde{r}_{mt} - \tilde{r}_{ft}] + \sum_{g=1}^{N} \beta_{I,n}^2[\tilde{r}_{mt} - \tilde{r}_{ft}]*ID_{gt} + \tilde{\varepsilon}_{it} \tag{13}$$

where $g = 1, 2, \cdots, N$ represents event number. To study the long-term effects of systematic risk, Eq (13) are re-estimated by making the individual dummy variables (*ID*) assume a value of zero before the event and one afterwards.

## 4. Data and empirical results

The daily data series downloaded from Thomson Reuters Eikon DataStream comprises the return index of 38 industries on the Vietnamese stock market between 2010 and 2020. We apply the MSCI Vietnam Index and Vietnam interbank 3-month rate as proxies for the market return and risk-free rate. Relating to the indicators for Fama and French's five-factor Asset Pricing model, we downloaded them from the Kenneth R. French data library at Dartmouth College.

Ten announcements surrounding the EVFTA from the first negotiation in 2010 until the ratification date in 2020 were collected from different sources (as displayed in Table 1). These announcements were collected from 04/10/2010 up to 12/02/2020. The first event was the Prime Minister of Vietnam and the President of the EU agreed to start negotiations on the EVFTA agreement. The latest event was when the European Parliament ratified EVFTA and EVIPA.

**Table 1. Events around the EVFTA.**

| Event | Date | Description |
|:---:|:---:|:---|
| 1 | 04/10/ 2010 | The Prime Minister of Vietnam and the President of the EU agreed to start negotiations on the EVFTA Agreement. |
| 2 | 26/06/ 2012 | Vietnam's Minister of Industry and Trade and EU Trade Commissioner officially launched negotiations for the EVFTA Agreement. |
| 3 | 02/12/ 2015 | Announcing the formal conclusion of negotiations for EVFTA. |
| 4 | 01/02/ 2016 | The preliminary text of the Agreement was officially announced. |
| 5 | 26/06/ 2018 | Vietnam and the EU have officially agreed to separate into two agreements: a free trade agreement (FTA) and an investment protection agreement (IPA). |
| 6 | 17/10/ 2018 | The European Commission has officially adopted EVFTA and IPA. |
| 7 | 25/06/ 2019 | The European Council approved the EVFTA and EVIPA deals. |
| 8 | 30/06/ 2019 | The European Union Vietnam Free Trade Agreement (EVFTA) was signed in Ha Noi. |
| 9 | 21/01/ 2020 | European Parliament Committee on International Trade approves a resolution asking for the European Parliament's ratification of the trade deals. |
| 10 | 12/02/ 2020 | European Parliament ratifies EVFTA and EVIPA. |

## 4.1 Immediate reaction analysis

Table 2 shows statistically significant abnormal returns of each sector following announcements around the European-Vietnam Free Trade Agreement (EVFTA). Overall, the EVFTA elicited significant responses from 20 out of 38 industries. Fourteen industries had an adverse reaction, and six sectors reacted positively. Besides, two of the ten events that were collected had no immediate impact on these industries (announcements 5 and 10). The fifth announcement was made on June 26, 2018, when Vietnam and the EU officially decided to split their deal into two parts: a free trade agreement (FTA) and an investment protection agreement (IPA). The European Parliament ratifies EVFTA and EVIPA on February 12, 2020, which is announcement 10.

While announcement 1 (made at the beginning of negotiations between the Prime Minister of Vietnam and the President of the European Union on October 4, 2010) had the strongest influence on all industries, all reactions were negative. On the other hand, announcement 9 (when there was the approval of EVFTA from the European Parliament Committee on February 21, 2020) resulted in positive reactions from several sectors, such as banks and financial services. Some other announcements, such as announcement 6 (when the European Commission officially adopted EVFTA and IPA on October 17, 2018), caused some contradicting reactions whereby some sectors, including mining and nonlife insurance, experienced positive reactions while the software services sector exhibited a negative response. From the economic perspective, the mining and nonlife insurance sectors will likely benefit when tariffs are removed, and the cost reduction could potentially make the products more competitive and result in higher profitability. However, the competition may become stiff for high-tech sectors such as software services, leading to lower profitability. From the investment perspective, this is an example of how the investors' perception of the announcement can lead to the differences between returns' reactions to the events. These results are relatively contradictory to the existing literature, where most studies only show positive impacts of FTAs on sectoral stock returns [5, 7]. The findings also suggest that FTAs might not benefit the entire economy, especially for a developing country like Vietnam.

**Table 2. Reactions following the events around the EVFTA (in %).**

| Sectors | Event | Date | AR | t-stat |
|---|---|---|---|---|
| *Negative Reactions* | | | | |
| Aerospace & Defense | 4 | 01/02/2016 | -10.58 | -3.19 |
| Automobiles & Parts | 1 | 04/10/2010 | -5.43 | -2.84 |
| Beverages | 3 | 02/12/2015 | -1.96 | -2.09 |
| Healthcare Equipment & Services | 3 | 02/12/2015 | -4.77 | -2.26 |
| Household Goods & Home Construction | 1 | 04/10/2010 | -3.54 | -2.17 |
| Industrial Engineering | 7 | 25/06/2015 | -1.13 | -2.25 |
| Industrial Transportation | 1 | 04/10/2010 | -2.92 | -2.08 |
| Media | 1 | 04/10/2010 | -4.64 | -2.21 |
| Oil and Gas Producer | 4 | 01/02/2016 | -1.52 | -2.07 |
| Real Estate Services | 1 | 04/10/2010 | -3.27 | -2.73 |
| Software Services | 6 | 17/10/2018 | -4.74 | -3.32 |
| Technology Hardware & Equipment | 1 | 04/10/2010 | -5.54 | -2.41 |
|  | 2 | 26/06/2012 | -2.73 | -2.09 |
| Tobacco | 2 | 26/06/2012 | -5.16 | -2.34 |
| Travel & Leisure | 1 | 04/10/2010 | -2.72 | -2.17 |
| *Positive Reactions* | | | | |
| Banks | 9 | 21/01/2020 | 2.75 | 3.70 |
| Financial Services | 9 | 21/01/2020 | 0.99 | 2.08 |
| Industrial Metals & Mining | 4 | 01/02/2016 | 1.95 | 2.30 |
| Leisure Goods | 8 | 30/06/2019 | 3.52 | 2.20 |
| Mining | 6 | 17/10/2018 | 1.38 | 2.31 |
| Nonlife Insurance | 6 | 17/10/2018 | 2.79 | 2.78 |

Amongst the sectors negatively affected, aerospace and defence experienced the highest negative reaction with an abnormal return of -10.58%, followed by technology hardware and equipment, and automobiles and parts sectors with abnormal returns of -5.54% and -5.43%, respectively. A plausible explanation for these adverse reactions is a consumption switch to foreign companies that usually provide better options and services for businesses because of the potential decrease in import tax and less stringent legislation.

After the European Parliament Committee on International Trade approved a resolution asking for the European Parliament's ratification of the trade deals (announcement 9), banks and the financial services sector exhibited positive abnormal returns of 2.75% and 0.99%, respectively. The results contradict those of [7], who found a negative (no) reaction of the banking (financial) sector to FTA announcements. FTAs are undoubtedly posing a challenge for domestic companies to drop their cost and enhance quality simultaneously. However, given Vietnam's comparative advantage in labour and material costs [53], the EVFTA creates opportunities for Vietnamese firms to reach a higher level of demand, hence possibly higher profitability. In addition, Vietnamese banks and financial services providers will also benefit from this free trade agreement since they can provide more capital or investment solutions not only to domestic firms but also to European ones. Our results are validated using several robustness tests, including the ranking test, the nonparametric conditional distribution test, market integration and the Fama-French five-factor model. Most of them are supported by at least one of these robustness tests, as shown in Table 3.

**Table 3. Robustness tests for the events around the EVFTA.**

| Sectors | Event | AR(%) | t-stat | tCorrado | Chesney | | Market Integration | | Fama French Five-Factor Model | |
|---|---|---|---|---|---|---|---|---|---|---|
| | | | | | CP | t-stat | AR(%) | t-stat | AR(%) | t-stat |
| **Negative Reactions** | | | | | | | | | | |
| Aerospace & Defense | 4 | -10.58 | -3.19 | -1.70 | 0.44 | 0.15 | -0.11 | -3.13 | -10.66 | -3.15 |
| Automobiles & Parts | 1 | -5.43 | -2.84 | -1.11 | N/A | N/A | -0.05 | -2.00 | -5.42 | -2.10 |
| Beverages | 3 | -1.96 | -2.09 | -1.72 | 0.07 | 1.73 | -0.02 | -1.48 | -2.26 | -2.20 |
| Healthcare Equipment & Services | 3 | -4.77 | -2.26 | -2.29 | 0.17 | 1.04 | -0.07 | -3.24 | -7.45 | -3.28 |
| Household Goods & Home Construction | 1 | -3.54 | -2.17 | -1.87 | N/A | N/A | -0.03 | -2.01 | -3.17 | -1.68 |
| Industrial Engineering | 7 | -1.13 | -2.25 | -0.41 | 0.01 | 3.38 | -0.01 | -1.29 | -1.08 | -2.09 |
| Industrial Transportation | 1 | -2.92 | -2.08 | -1.89 | N/A | N/A | -0.03 | -0.03 | -3.54 | -1.71 |
| Media | 1 | -4.64 | -2.21 | -2.13 | N/A | N/A | -0.05 | -2.47 | -5.11 | -2.06 |
| Oil & Gas Producers | 4 | -1.52 | -2.07 | -1.07 | 0.33 | 0.47 | -0.01 | -1.56 | -1.85 | -2.30 |
| Real Estate Services | 1 | -3.27 | -2.73 | -2.13 | N/A | N/A | -0.02 | -2.30 | -2.70 | -1.56 |
| Software Services | 6 | -4.74 | -3.32 | -1.97 | 0.23 | 0.79 | -0.05 | -2.89 | -4.70 | -3.09 |
| Technology Hardware & Equipment | 1 | -5.54 | -2.41 | -2.30 | N/A | N/A | -0.05 | -2.30 | -5.50 | -2.00 |
| | 2 | -2.73 | -2.09 | -1.81 | 0.22 | 0.85 | -0.02 | -1.58 | -3.54 | -2.01 |
| Tobacco | 2 | -5.16 | -2.34 | -2.15 | 0.20 | 0.93 | -0.05 | -2.23 | -5.63 | -2.44 |
| Travel & Leisure | 1 | -2.72 | -2.17 | -2.06 | N/A | N/A | -0.01 | -0.56 | -1.18 | -0.80 |
| **Positive Reactions** | | | | | | | | | | |
| Banks | 9 | 2.75 | 3.70 | 2.33 | 0.09 | 1.52 | 0.04 | 4.11 | 1.41 | 1.71 |
| Financial Services | 9 | 0.99 | 2.08 | 0.11 | 0.01 | 3.78 | 0.03 | 3.19 | 0.56 | 1.10 |
| Industrial Metals & Mining | 4 | 1.95 | 2.30 | 0.29 | 0.45 | 0.13 | 0.02 | 2.24 | 1.69 | 1.82 |
| Leisure Goods | 8 | 3.52 | 2.20 | -0.38 | 0.24 | 0.75 | 0.00 | 0.03 | 3.15 | 1.86 |
| Mining | 6 | 1.38 | 2.31 | 0.54 | 0.47 | 0.07 | 0.01 | 1.28 | 1.38 | 1.87 |
| Nonlife Insurance | 6 | 2.79 | 2.78 | 2.06 | 0.03 | 2.33 | 0.03 | 2.28 | 2.82 | 2.36 |

## 4.2 Early and delayed reaction analysis

The efficient market hypothesis (EMH) assumes that the stock market reacts immediately to any available public information. Therefore, abnormal returns are only worthy of consideration on the first day of the announcement. Information, however, can also be leaked before the intended announcement date. To capture this phenomenon, the market anticipation of five days and two days before the announcements of EVFTA (as measured by the cumulative abnormal return, CAR(-5) and CAR(-2) respectively) using Eqs (4) and (5) (see Table 4). In general, our results show that eight sectors experienced positive CAR (-5) or CAR (-2), whereas one and four sectors experienced mixed and negative outcomes, respectively. We observe that energy-intensive sectors such as industrial metals and mining reacted unfavourably to the news. A potential explanation for this reaction is that these sectors expect convergence of Vietnamese standards with those of the EU according to the provisions under EVFTA. It means the polluting Vietnamese firms will need to gradually follow EU standards such as sustainable development and environmentally friendly production. To this extent, domestic companies are encouraged to utilise renewable energy and improve their stance on corporate social responsibility to comply with the EU parties' requirements.

On the contrary, several key export sectors to Europe, including electronic and electrical equipment, oil equipment and services, experienced positive reactions to the news of EVFTA. Electronic and electrical equipment, and oil equipment and services, for instance, recorded high positive cumulative responses with a CAR(-5) of 10.49% (with a t-statistic of 2.35) and

**Table 4. Market anticipation following the events around EVFTA (in %).**

| Sectors | Event | CAPM | | | | Fama French Five-Factor Model | | | |
|---|---|---|---|---|---|---|---|---|---|
| | | CAR(-5) | t-stat | CAR(-2) | t-stat | CAR(-5) | t-stat | CAR(-2) | t-stat |
| **Mixed Reactions** | | | | | | | | | |
| Industrial Engineering | 5 | -0.09 | -0.08 | 1.65 | 2.28 | 0.48 | 0.41 | 1.46 | 1.84 |
| **Negative Reactions** | | | | | | | | | |
| Alternative Energy | 8 | -2.59 | -0.94 | -3.43 | -2.02 | -3.90 | -1.27 | -4.01 | -2.05 |
| Industrial Metals & Mining | 6 | -4.16 | -2.50 | -1.35 | -1.37 | -3.04 | -1.73 | -1.64 | -1.42 |
| Tobacco | 7 | -5.72 | -1.96 | -0.11 | -0.06 | -6.36 | -2.10 | -1.13 | -0.58 |
| Technology Hardware & Equipment | 4 | -2.48 | -1.61 | -2.76 | -2.67 | -1.37 | -0.83 | -1.94 | -1.72 |
| **Positive Reactions** | | | | | | | | | |
| Electronic & Electrical Equipment | 2 | 6.38 | 1.52 | 7.04 | 2.72 | 8.78 | 2.35 | 2.56 | 1.12 |
| Gas, Water & Multiutilities | 4 | 5.19 | 2.17 | 1.26 | 0.80 | 4.87 | 2.12 | 1.18 | 0.83 |
| Healthcare Equipment & Services | 3 | 3.38 | 1.97 | 1.84 | 1.52 | 5.34 | 0.92 | -0.90 | -0.26 |
| Leisure Goods | 4 | 11.00 | 2.07 | 3.90 | 1.20 | 10.75 | 2.11 | 7.40 | 2.33 |
| Non-equity Investment Trust | 5 | 2.19 | 2.09 | 1.60 | 2.18 | 7.97 | 3.10 | 3.25 | 1.95 |
| Oil Equipment & Services | 4 | 5.11 | 2.27 | 1.37 | 0.96 | 11.21 | 2.74 | 4.29 | 1.63 |
| | 5 | 1.79 | 1.09 | 2.39 | 2.27 | 5.25 | 1.42 | 4.18 | 1.69 |
| Software Services | 6 | 4.59 | 2.53 | 2.06 | 1.90 | 5.91 | 1.83 | 4.22 | 2.02 |
| Support Services | 6 | 3.83 | 2.01 | 0.72 | 0.66 | 3.68 | 2.51 | 2.24 | 2.22 |

8.81% (with a t-statistic of 2.57), respectively. These findings imply that several key Vietnamese sectors expect the benefits of EVFTA to outweigh the costs in their respective sectors, and hence, the investors treat this as favourable news. Table 4 also displays the robustness test results for cumulative abnormal returns two and five days before the event dates using the Fama French five-factor model as an alternative asset pricing model.

Nevertheless, behavioural finance posits that market participants joining the financial market with a representative bias could unavoidably encounter under- (over) responses to the latest information, whereby market participants are likely to respond continuously to any new messages in the market. Following previous studies in the literature, we capture these reactions by estimating cumulative abnormal returns two days, five days, and ten days after the events. Interestingly, we document seven sectors experiencing negative delayed responses (see Table 5), whereas four and ten sectors exhibited positive and mixed outcomes, respectively, as shown in Table 6. The announcement on event 10 (when the European Parliament ratifies

**Table 5. Negative delayed reaction following the events around EVFTA (in %).**

| Sectors | Event | CAPM | | | | | | Fama French Five-Factor Model | | | | | |
|---|---|---|---|---|---|---|---|---|---|---|---|---|---|
| | | CAR2 | t-stat | CAR5 | t-stat | CAR10 | t-stat | CAR2 | t-stat | CAR5 | t-stat | CAR10 | t-stat |
| Chemicals | 8 | -1.77 | -2.16 | -2.36 | -1.85 | -4.85 | -2.74 | -2.22 | -2.41 | -2.80 | -1.95 | -4.99 | -2.50 |
| Electronic & Electrical Equipment | 9 | -1.42 | -0.89 | -1.49 | -0.58 | -7.72 | -2.22 | -1.48 | -0.95 | -1.81 | -0.70 | -5.94 | -1.70 |
| Food & Drugs Retailers | 10 | -4.61 | -1.38 | -11.87 | -2.52 | -17.24 | -2.78 | -5.13 | -1.33 | -12.39 | -2.32 | -16.67 | -2.59 |
| Gas, Water & Multiutilities | 3 | -2.61 | -1.94 | -0.06 | -0.03 | -6.95 | -2.2 | -2.45 | -1.73 | 0.05 | 0.02 | -6.38 | -1.88 |
| | 10 | -1.11 | -1.31 | -3.15 | -2.36 | -5.34 | -2.52 | -1.9 | -1.89 | -4.51 | -2.75 | -5.26 | -2.02 |
| Household Goods & Home Construction | 10 | -0.89 | -0.81 | -1.32 | -0.72 | -6.92 | -2.55 | -1.4 | -1.31 | -2.02 | -1.18 | -4.44 | -1.81 |
| Non-equity Investment Trust | 8 | -0.45 | -0.46 | -0.06 | -0.04 | -7.72 | -3.28 | -0.87 | -0.71 | -0.48 | -0.24 | -7.29 | -2.78 |
| | 9 | -1.44 | -1.7 | -1.28 | -0.93 | -3.93 | -1.97 | -1.29 | -1.45 | -1.31 | -0.86 | -4.44 | -1.99 |
| Oil Equipment & Services | 4 | -4.31 | -1.97 | -2.47 | -0.74 | -2.17 | -0.47 | -3.8 | -1.44 | -1.02 | -0.25 | -0.88 | -0.16 |

**Table 6. Positive and mixed delayed reaction following the events around the agreement (in %).**

| Sectors | Event | CAPM | | | | | | Fama French Five-Factor Model | | | | | |
|---|---|---|---|---|---|---|---|---|---|---|---|---|---|
| | | CAR2 | t-stat | CAR5 | t-stat | CAR10 | t-stat | CAR2 | t-stat | CAR5 | t-stat | CAR10 | t-stat |
| **Positive Reactions** | | | | | | | | | | | | | |
| Construction | 6 | 1.49 | 2.51 | 0.8 | 0.86 | 1.28 | 0.95 | 1.56 | 1.99 | 0.87 | 0.71 | 2.9 | 1.59 |
| Nonlife Insurance | 5 | 2.42 | 2 | 1.31 | 0.83 | 1.54 | 0.83 | 3.56 | 2.56 | 0.59 | 0.32 | 1.81 | 0.82 |
| Personal Goods | 9 | 2.03 | 2.3 | 1.98 | 1.41 | 0.3 | 0.15 | 1.81 | 2.03 | 1.69 | 1.27 | 0.87 | 0.48 |
| Support Services | 10 | 2.85 | 2.71 | 2.52 | 1.42 | 5.84 | 2.33 | 1.41 | 1.84 | 0.62 | 0.49 | 2.13 | 1.18 |
| **Mixed Reactions** | | | | | | | | | | | | | |
| Aerospace & Defense | 3 | -0.22 | -0.06 | 11.23 | 2.28 | 4.06 | 0.73 | 0.12 | 0.03 | 11.29 | 2.24 | 5.03 | 0.85 |
| Beverages | 9 | 0.12 | 0.13 | 0.06 | 0.04 | -4.69 | -2.42 | 0.07 | 0.07 | -0.45 | -0.28 | -5.72 | -2.42 |
| Food Producers | 8 | 0.68 | 1.34 | 1.75 | 1.97 | 1.8 | 1.45 | 0.6 | 0.98 | 1.77 | 1.81 | 1.76 | 1.35 |
| | 9 | 0.22 | 0.41 | 0.34 | 0.37 | -3.01 | -2.4 | -0.09 | -0.16 | -0.22 | -0.22 | -3.28 | -2.27 |
| General Retailers | 3 | 2.91 | 2.53 | 5.35 | 2.82 | 7.38 | 2.66 | 1.02 | 0.82 | 5.21 | 2.52 | 6.00 | 1.96 |
| | 8 | -2.14 | -2.75 | -1.18 | -0.94 | -1.72 | -0.97 | -1.23 | -1.44 | -0.4 | -0.29 | -0.65 | -0.33 |
| Industrial Transportation | 7 | 1.01 | 2.07 | 0.95 | 1.31 | -0.49 | -0.5 | 0.97 | 1.63 | 0.99 | 1.17 | -0.42 | -0.38 |
| Leisure Goods | 6 | -7.7 | -3.67 | -7.89 | -2.74 | -6.63 | -1.79 | -9.44 | -3.72 | -8.07 | -2.35 | -6.99 | -1.54 |
| | 7 | 4.07 | 1.92 | 6.29 | 2.12 | 0.61 | 0.16 | 4.1 | 1.82 | 6.04 | 1.85 | 0.83 | 0.20 |
| Media | 5 | 2.35 | 2.41 | -0.85 | -0.49 | 0.2 | 0.09 | 3.33 | 3.35 | 1.73 | 0.91 | 2.69 | 1.03 |
| Oil & Gas Producers | 4 | -1.97 | -1.96 | 0.99 | 0.64 | 2.02 | 1.01 | -1.93 | -1.74 | 1.56 | 0.9 | 2.36 | 0.96 |
| | 6 | 3 | 2.26 | 1.69 | 0.83 | 4.4 | 1.64 | 3.24 | 2.38 | 1.8 | 0.9 | 4.85 | 1.91 |
| | 10 | -3.08 | -2.58 | -4.51 | -2.26 | -8.54 | -2.97 | -3.59 | -2.98 | -5.24 | -2.6 | -7.66 | -2.67 |
| Software Services | 9 | 3.74 | 2.00 | 3.63 | 1.38 | -4.46 | -1.27 | 3.56 | 1.81 | 3.14 | 1.10 | -4.42 | -1.09 |
| Travel & Leisure | 10 | 1.53 | 2.03 | -0.94 | -0.73 | -0.66 | -0.35 | 1.18 | 1.48 | -1.68 | -1.26 | -0.74 | -0.38 |

EVFTA and EVIPA), for instance, affects the food and drug retailers negatively, with CAR(5) of -11.87% (with a t-statistic of -2.52) and CAR(10) of -17.24 (with a t-statistic of -2.78). Another example is the oil equipment sector, which exhibited a statistically significant negative reaction 2 days following announcement 4 when the preliminary text of the Agreement was officially announced. However, the oil equipment and services sector reacted positively 5 days prior to this announcement with a CAR (-5) of 5.11% (with a t-statistic of 2.27). Therefore, this is a typical example to show the oil equipment services sector's ability to pass the cost to their customers. Tables 5 and 6 also indicate the robustness test for these cumulative delayed abnormal returns to the EVFTA-related announcements using the Fama French five-factor model.

## 4.3 Risk structure analysis

Apart from investigating the return, the estimation of risks around the Free Trade Agreement is also paramount. Thus, to address how the risk structure changes following these events, we examine the effects of 10 announcements related to EVFTA on the short-term and long-term systematic risk of 38 sectors. Table 7 presents the industries experiencing changes in aggregate systematic risk by estimating the intercepts, the betas before the events, and the aggregated changes in betas thereafter. Only two sectors in the Vietnamese stock market possessed statistically significant aggregate changes in systematic risk. Aerospace and Defense, for instance, is the most influenced with a sharp rise of 0.62 to 1.39 (with a t-statistic of 2.58); the follower is tobacco, which experienced a gradual increase of 0.54 (with a t-statistic of 2.16). These outcomes are formed on the GARCH model and supported by other approaches such as TARCH, EGARCH, and PARCH specifications (Results are not reported for brevity purpose and

**Table 7. Aggregate change in systematic risk following the announcements around EVFTA.**

| Sectors | Intercept | t-stat | Beta | t-stat | Aggregate Change in Beta | t-stat |
|---|---|---|---|---|---|---|
| Aerospace & Defense | 0 | 0.35 | 0.77 | 29.06 | 0.62 | 2.58 |
| Alternative Energy | 0 | 0.67 | 0.75 | 38.96 | -0.33 | -0.23 |
| Automobiles & Parts | 0 | -1.04 | 0.95 | 72.5 | -0.07 | -0.30 |
| Banks | 0 | -1.72 | 0.98 | 179.68 | 0.04 | 0.51 |
| Beverages | 0 | 1.09 | 0.79 | 81.16 | 0.14 | 0.78 |
| Chemicals | 0 | -1.10 | 0.86 | 115.59 | -0.01 | -0.02 |
| Construction & Materials | 0 | -2.47 | 0.89 | 181.13 | 0.06 | 0.62 |
| Electronic & Electrical Equipment | 0 | -0.61 | 0.82 | 49.36 | -0.19 | -0.63 |
| Electricity | 0 | 1.68 | 0.86 | 131.53 | -0.07 | -0.28 |
| Financial Services | 0 | -3.27 | 0.96 | 143.5 | -0.01 | -0.11 |
| Fixed Line Telecommunications | 0 | -1.17 | 0.94 | 68.13 | 0.18 | 0.75 |
| Food & Drug Retailers | 0 | -0.53 | 0.55 | 13.67 | 0.07 | 0.07 |
| Food Producers | 0 | -1.61 | 0.89 | 173.8 | 0.17 | 1.77 |
| Forestry & Papers | 0 | -0.53 | 0.91 | 68.98 | 0.09 | 0.37 |
| Gas, Water & Multiutilities | 0 | 0.14 | 0.82 | 110.15 | 0.12 | 0.50 |
| General Industrials | 0 | 0.09 | 0.86 | 140.33 | 0.03 | 0.34 |
| General Retailers | 0 | 0.21 | 0.84 | 124.78 | -0.04 | -0.22 |
| Healthcare Equipment & Services | 0 | 0.04 | 0.84 | 44.42 | -0.01 | -0.02 |
| Household Goods & Home Construction | 0 | -2.16 | 0.90 | 124.00 | 0.14 | 0.49 |
| Industrial Engineering | 0 | -1.01 | 0.84 | 127.82 | 0.02 | 0.11 |
| Industrial Metals & Mining | 0 | -3.26 | 0.90 | 118.29 | -0.06 | -0.77 |
| Industrial Transportation | 0 | -1.60 | 0.88 | 151.54 | -0.02 | -0.06 |
| Leisure Goods | 0 | -2.16 | 0.76 | 52.71 | 0.24 | 0.39 |
| Media | 0 | 0.77 | 0.81 | 84.16 | -0.05 | -0.17 |
| Mining | 0 | -3.08 | 0.91 | 116.68 | 0.18 | 0.99 |
| Non-equity Investment Trust | 0 | 0.12 | 0.84 | 113.32 | -0.05 | -0.13 |
| Nonlife Insurance | 0 | 0.76 | 0.94 | 96.58 | 0.26 | 0.94 |
| Oil Equipment & Services | 0 | -2.93 | 1.11 | 100.49 | 0.06 | 0.09 |
| Oil & Gas Producers | 0 | -0.02 | 0.85 | 110.81 | 0.17 | 0.72 |
| Personal Goods | 0 | -0.29 | 0.82 | 97.32 | -0.02 | -0.06 |
| Pharmaceuticals & Biotechnology | 0 | 0.98 | 0.78 | 114.94 | -0.05 | -0.28 |
| Real Estate Investment & Services | 0 | -2.00 | 0.96 | 182.21 | 0.04 | 0.42 |
| Real Estate Investment Trust | 0 | -0.99 | 0.69 | 84.49 | 0.17 | 0.52 |
| Software & Computer Services | 0 | -2.17 | 0.87 | 74.10 | 0.15 | 0.72 |
| Support Services | 0 | -0.79 | 0.82 | 111.15 | -0.03 | -0.06 |
| Technology Hardware & Equipment | 0 | -0.59 | 0.88 | 91.57 | 0.11 | 1.47 |
| Tobacco | 0 | 0.71 | 0.82 | 57.48 | 0.54 | 2.16 |
| Travel & Leisure | 0 | -0.95 | 0.78 | 114.28 | 0.03 | 0.03 |

available upon request). These results are consistent with those of [19], who found that the country with a lower economic power in FTAs tend to experience an increase in stock market volatility.

However, most industries did not show any changes in systematic risk because the results of 10 announcements could cancel out each other. Therefore, to solve this issue, Eq (13) is proposed to inspect short-term changes in systematic risks, and Fig 2 visualises the changes in 38 industries' beta for the period 2000–2020. There was a relatively stable systematic risk of these sectors in the Vietnamese stock market until the first diamond risk structure appeared around

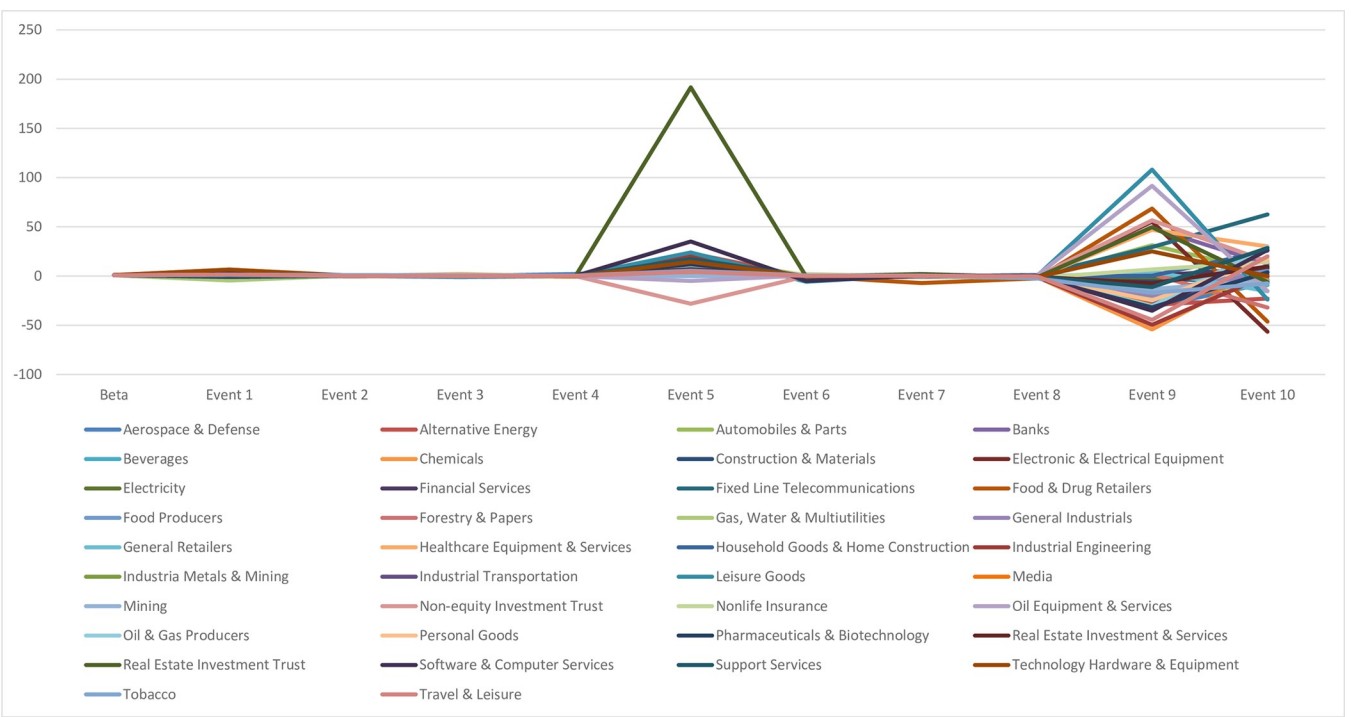

**Fig 2. Short-term changes in systematic risk following EVFTA announcements.**

announcement 5 when Vietnam and the EU officially agreed to separate into two agreements, including EVFTA and IPA, on June 26, 2018. The EVFTA and IPA were included in a single text and were to be ratified by the EU and Vietnam without involving the Member States initially. However, learning from the experience of the EU-Singapore FTA, whereby non-direct investment and investor-state dispute settlement mechanisms are shared competencies, on which the EU shares decision-making powers with the Member States, Vietnam and the EU decided to split the deal into two agreements to speed up the ratification process of the EVFTA which could be concluded by the EU alone without involving the EU Member States. This decision gave a strong signal to the market that the FTA could come into force earlier than expected, creating short-term market volatility. For instance, real estate investment trusts experienced a sharp rise in short-term systematic risk when the agreements were officially agreed to separate into two (announcement 5) and a decrease in short-term systematic risk when the European Commission officially adopted these agreements on the next event day (announcement 6). Then, the second diamond risk structure was detected around announcement 9 when the European Parliament Committee on International Trade approved a resolution on 21st January 2020, implying a high level of uncertainty.

Regarding the long-term systematic risk, Fig 3 displays that the announcements around the Free Trade Agreement affect long-term systematic risks, which differ sector by sector. It can be seen that changes in systematic risk have many significant variations of beta in industrial effects from one industry to another industry. Still, there are two diamond risk formulations around announcements 3 and 7. A more apparent diamond risk structure was formed after announcement 7 when the European Council approved the EVFTA and IPA deals. The EVFTA and IPA signed with Vietnam were the most ambitious deals of their type ever concluded by the EU and a developing country; hence, this announcement was a significant landmark for Vietnam and could potentially change the Vietnam economy in the long term. The

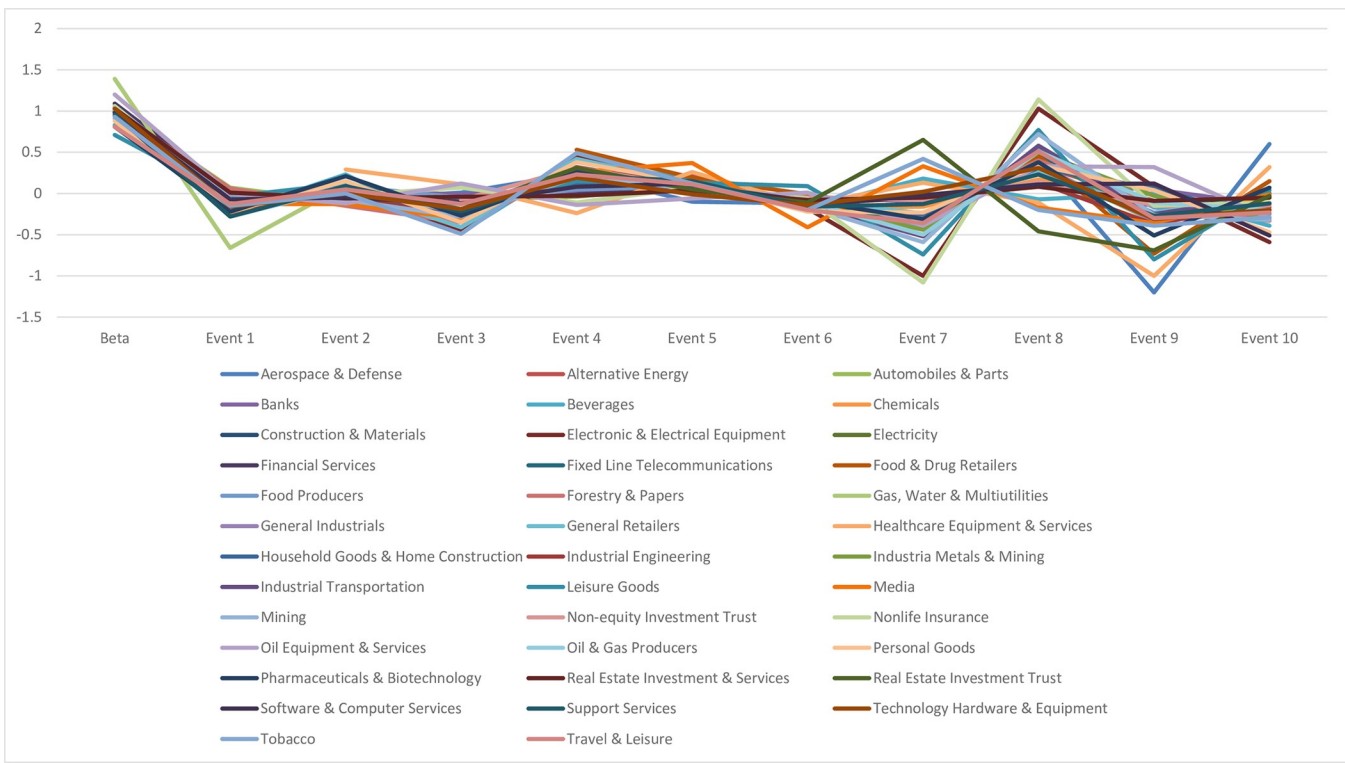

**Fig 3. Long-term changes in systematic risk following EVFTA announcements.**

findings have also shown that ten events around the Free Trade Agreement could cause short- and long-term uncertainty in the Vietnamese stock market.

## 5. Conclusion

The European Union and Vietnam Free Trade Agreement (EVFTA), signed in mid-2019, is expected to contribute significantly to the Vietnamese economy. This study examines Vietnam's stock market reaction to the events related to the EVFTA. For this purpose, we measure abnormal returns around the news events using established event methodologies and several ARCH-type additional tests for checking the robustness of the results. We measure immediate, early and delayed reactions before and after the announcements and find several interesting results.

First, Vietnam stock markets show strong and statistically significant sensitivity to the EVFTA-related events immediately before the announcements. Second, the news events negatively affect some industry sectors, exhibiting an unfavourable response from the investing community. The EVFTA causes consumption to switch to imported brands, resulting in lower demand for local brands. However, industry sectors that enjoy a lower cost of production exhibit a positive response to the events. Third, sectoral performance around the events shows an interesting picture. A varying degree of impact on the sectors is attributable to the industry characteristics, changes in the level of competition owing to the introduction of new tariff structures, and new entrants to the market. The EVFTA introduces a new level of market risk, affecting all industries, which can impact investor risk premiums. Fourth, most industry sectors show positive cumulative abnormal returns five or two days before announcements. After the announcements, cumulative abnormal returns are mixed. Finally, our results are robust

due to the use of established event study methodology and various asset pricing models. In addition, we observe two diamond risk structures when there are certain major changes in the FTA between the EU and VN, implying that the Vietnamese sectors view these changes differently and experience opposite changes in systematic risks. These results also show that not all sectors favour an FTA, whereby the local companies might face a higher level of competition from foreign firms. The findings are robust after several robustness tests using various ARCH-type models.

The findings of this study have several significant implications for investors and policymakers. First, the investors can potentially take advantage of future FTAs between the EU and other developing countries, whereby they can increase their allocation in the sectors that benefit from the FTAs and reduce or eliminate the allocation in the sectors that may experience an increase in risk due to the FTAs. Second, policymakers can utilise the empirical evidence from this study to make appropriate policy adjustments targeting specific sectors that may be more vulnerable to the FTAs and mitigate the potential adverse impacts on these sectors.

This study is not without limitations, whereby the results might not be applicable in the context of other countries. In addition, although the study tries to isolate the impact of the EVFTA-related events on the Vietnam stock market using various asset pricing models and robustness tests, the results might suffer from the effects of co-founding events around the sample period.

## Author Contributions

**Conceptualization:** Huy Pham.

**Data curation:** Mai Bui.

**Funding acquisition:** Tuan Hung Vu.

**Investigation:** Vikash Ramiah.

**Methodology:** Mai Bui, Van Nguyen.

**Resources:** Tuan Hung Vu.

**Software:** Van Nguyen.

**Supervision:** Huy Pham.

**Validation:** Tuan Chu, Tuan Hung Vu.

**Visualization:** Tuan Chu.

**Writing – original draft:** Priyantha Mudalige, Hanh Le.

**Writing – review & editing:** Huy Pham, Vikash Ramiah.

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
