## [Decision Letter · Decision Letter 0]

22 Aug 2023

PONE-D-23-21666The Effects of Free Trade Agreements on the Stock Market: Evidence from VietnamPLOS ONE

Dear Dr. VU,

Thank you for submitting your manuscript to PLOS ONE. After careful consideration, we feel that it has merit but does not fully meet PLOS ONE’s publication criteria as it currently stands. Therefore, we invite you to submit a revised version of the manuscript that addresses the points raised during the review process.

We look forward to receiving your revised manuscript.

Kind regards,

Ricky Chee Jiun Chia

Academic Editor

PLOS ONE

Reviewers' comments:

Reviewer's Responses to Questions

**Comments to the Author**

1. Is the manuscript technically sound, and do the data support the conclusions?

Reviewer #1: Partly

Reviewer #2: Partly

2. Has the statistical analysis been performed appropriately and rigorously? 

Reviewer #1: I Don't Know

Reviewer #2: Yes

3. Have the authors made all data underlying the findings in their manuscript fully available?

Reviewer #1: Yes

Reviewer #2: No

4. Is the manuscript presented in an intelligible fashion and written in standard English?

Reviewer #1: Yes

Reviewer #2: Yes

5. Review Comments to the Author

Reviewer #1: This paper discusses the effects of Free Trade Agreements on the Vietnam stock market. Manuscripts describe, in part, technically sound scientific research and provide data to support conclusions.It is rich in content, but there are still some problems.

Reviewer #2: The manuscript is interesting, although the topic is not new, at least from the perspective of investigating the effects of trade agreements on industries and stock markets. However, it is argued that one of the contributions of this manuscript resides in the analysis of these effects for a developing economy, Vietnam, that has recently signed a free-trade agreement with the EU. The manuscript needs several changes before being published.

1. The Introduction should stress more the originality and contributions to the literature, particularly since the topic is not new.

2. The Literature review needs organisation and structure. As it is, it looks amalgamated and constructed from various paragraphs that do not logically follow one after the other. Also, i encourage the authors to use the literature review to build their research questions and hypotheses, by highlighting the research gap.

3. The methodology is well explained, but the low number of events included in th study (10) might impact the findings.

4. The results need to be better explained, particularly the differences between returns' reaction to each of the 10 events, as well as those between the economic sectors.

5. There is no discussion on the results, the authors do not contrast their findings against other studies.

6. Conclusions need to highlight the implications of the findings, as well as to present the limitations of the study and the directions for future research that might be identified based on the results.

6. PLOS authors have the option to publish the peer review history of their article (what does this mean?). If published, this will include your full peer review and any attached files.

Reviewer #1: No

Reviewer #2: No

---

## [Author Response · Author response to Decision Letter 0]

6 Oct 2023

Dear Professor Hung Do,

On behalf of the co-authors, I would like to thank you in advance for reconsidering and handling our research article. We have made the adjustments to our paper following the reviewers’ comments.

Overall, this research makes the following contributions to the literature. First, we deepen the understanding of informed trading by providing crucial insights into the stock market reaction around the EVFTA announcements on the Vietnam stock market. Second, we contribute to the limited literature on risk structure caused by free trade agreements by providing a comprehensive risk analysis. 

In general, our findings are consistent with those of Hanson and Song (1998), who examined the effects of the North American Free Trade Agreement on the U.S. and Mexico stock markets and found mixed reactions to the free trade agreement announcements. The results, however, contradict those of Parinduri and Thangavelu (2012), who examined the effects of the U.S. ad Singapore Free Trade Agreement on the Singapore stock market and found the FTA increased the value of firms. A plausible explanation for this contradiction is the higher degree of uncertainty that EVFTA brings to Vietnamese firms, whereby the EU consists of several countries, and export (import) activities might be far more complicated than trading with a single country. 

Regarding the data availability, the data used in this study was collected from Refinitiv Eikon Datastream, hence we are not able to publicly share the data without Refinitiv’s permission. The data request might be sent to Refinitiv for further information.

Sincerely yours,

Tuan Vu

---

## [Editor Report · Decision Letter 1]

2 Nov 2023

The Effects of Free Trade Agreements on the Stock Market: Evidence from Vietnam

PONE-D-23-21666R1

Dear Dr. Tuan Hung Vu,

We’re pleased to inform you that your manuscript has been judged scientifically suitable for publication and will be formally accepted for publication once it meets all outstanding technical requirements.

Kind regards,

Ricky Chee Jiun Chia

Academic Editor

PLOS ONE
---

## [Editor Report · Acceptance letter]

10 Nov 2023

PONE-D-23-21666R1 

The effects of free trade agreements on the stock market: Evidence from Vietnam 

Dear Dr. Vu:

I'm pleased to inform you that your manuscript has been deemed suitable for publication in PLOS ONE. Congratulations! Your manuscript is now with our production department. 

Kind regards, 

on behalf of

Dr. Ricky Chee Jiun Chia 

Academic Editor

PLOS ONE